# The Novel Role of Noncoding RNAs in Modulating Platelet Function: Implications in Activation and Aggregation

**DOI:** 10.3390/ijms24087650

**Published:** 2023-04-21

**Authors:** Giovanni Cimmino, Stefano Conte, Domenico Palumbo, Simona Sperlongano, Michele Torella, Alessandro Della Corte, Paolo Golino

**Affiliations:** 1Department of Translational Medical Sciences, Section of Cardiology, University of Campania Luigi Vanvitelli, L. Bianchi Street, 80131 Naples, Italyalessandro.dellacorte@unicampania.it (A.D.C.);; 2Cardiology Unit, Azienda Ospedaliera Universitaria Luigi Vanvitelli, Piazza Miraglia, 80138 Naples, Italy; 3Department of Translational Medical Sciences, Section of Lung Diseases, University of Campania Luigi Vanvitelli, L. Bianchi Street, 80131 Naples, Italy

**Keywords:** platelet activation, platelet transcriptome, platelet proteome, thrombosis

## Abstract

It is currently believed that plaque complication, with the consequent superimposed thrombosis, is a key factor in the clinical occurrence of acute coronary syndromes (ACSs). Platelets are major players in this process. Despite the considerable progress made by the new antithrombotic strategies (P2Y12 receptor inhibitors, new oral anticoagulants, thrombin direct inhibitors, etc.) in terms of a reduction in major cardiovascular events, a significant number of patients with previous ACSs treated with these drugs continue to experience events, indicating that the mechanisms of platelet remain largely unknown. In the last decade, our knowledge of platelet pathophysiology has improved. It has been reported that, in response to physiological and pathological stimuli, platelet activation is accompanied by de novo protein synthesis, through a rapid and particularly well-regulated translation of resident mRNAs of megakaryocytic derivation. Although the platelets are anucleate, they indeed contain an important fraction of mRNAs that can be quickly used for protein synthesis following their activation. A better understanding of the pathophysiology of platelet activation and the interaction with the main cellular components of the vascular wall will open up new perspectives in the treatment of the majority of thrombotic disorders, such as ACSs, stroke, and peripheral artery diseases before and after the acute event. In the present review, we will discuss the novel role of noncoding RNAs in modulating platelet function, highlighting the possible implications in activation and aggregation.

## 1. Introduction

The rupture (or the erosion) of an atherosclerotic plaque with a superimposed platelet aggregation is the most accepted cause of acute coronary syndromes (ACSs) [1].

Unfortunately, the vulnerable plaques cannot be identified in everyday clinical practice, and new molecular biomarkers are desirable to allow a more efficient estimation of the risk of the most encountered complications of atherothrombosis, such as ACSs and cardiovascular deaths [2,3].

Platelet activation is a key element in the genesis of clinical complications of atherosclerotic coronary diseases. Indeed, the availability of increasingly efficient antiplatelet drugs had a significant impact on the prognosis of patients with ACSs [4,5,6]. Moreover, the presence of high on-treatment platelet reactivity is an important predictor of major adverse cardiovascular events (MACEs) in patients with chronic ischemic heart disease [7,8,9]. The presence of type 2 diabetes mellitus (DM2) or chronic kidney diseases (CDs) also has a significant impact on platelet activation [10,11] and represents a prognostic factor with an important additive value [12]. In addition, the presence of peripheral arterial disease (PAD) of the lower limbs and/or peripheral ischemia is associated with an alteration in the responsiveness of the platelets to shear stress and to pharmacological inhibition [13,14]. However, despite the considerable progress made by new antithrombotic and anticoagulant therapies (P2Y12 receptor inhibitors, new oral anticoagulants, thrombin direct inhibitors, etc.) in terms of MACE reduction, a significant number of patients with previous MACE treated with these drugs continue to experience events [15], indicating that the mechanisms of platelet activation in these patients are still largely unknown. For years, platelets have been considered only the final effector of the coagulation cascade mainly involved in thrombus formation [16]. However, in the last decade, great efforts have been directed toward a better understanding of platelet pathophysiology and discovering its functions beyond aggregation, such as immune modulation [17]. 

It has been found that, in response to physiological and pathological stimuli, platelet activation is associated with de novo protein synthesis, through a rapid and particularly well-regulated translation of resident mRNAs of megakaryocytic derivation [18]. Although platelets are anucleate, they indeed contain an important fraction of mRNAs that can be quickly used for protein synthesis following their activation [17,18]. However, the mechanisms underlying the modulation of the activity of these mRNAs are still not completely known. 

Moreover, beyond their well-established indispensable role in regulating hemostasis, platelets are increasingly emerging as key regulators in several processes, including inflammatory and immune pathways, response to both viral and bacterial infections, cancer, vascular and lymphatic development, the maintenance of vascular integrity, the maturation of the circulatory system, the formation of new blood vessels, wound healing, and bone formation [19]. This extraordinary ability of platelets to regulate so many physiological and pathophysiological conditions depends on their biochemical and functional heterogeneity, and on their capacity to store and release a wide range of biologically active substances via their granules and microparticles [17,20,21,22]. Platelets also secrete a large number of noncoding RNAs (miRNAs, lncRNAs, circRNAs, or snoRNAs) [23,24].

In the present review, the emerging regulatory pathways involved in platelet activation and aggregation will be discussed. 

## 2. The Classic View of Platelets in Thrombosis and Hemostasis

Platelets are the final effectors of the hemostatic process with the primary role to prevent blood loss if vascular damage occurs [16]. This process is highly regulated and involves vessels, vascular wall components, platelets, the coagulation cascade, and the fibrinolytic system [16,17]. Schematically, the hemostatic process may be divided into four interconnected steps [25]: (1) vascular phase: In the beginning, there is a short period of vasoconstriction (due to reflex neurogenic mechanisms and humoral factors such as endothelin, which is a potent vasoconstrictor of endothelial origin). Vascular contraction is more evident in vessels with well-defined muscle walls and serves to reduce momentary blood loss; (2) platelet phase: Endothelial cell injury exposes the subendothelial thrombogenic materials, to which platelets adhere, thus entering an “activation” state [16,26]. At this moment, a change in platelet shape and an exocytosis reaction occur [27]. Then, platelets release different factors from their own granules (ADP, TXA2, serotonin, and others) that recruit further platelets, leading to the final plug formation [16]. This reaction occurs within a few minutes of the injury and constitutes the so-called primary hemostasis [25]. In the case of capillary lesions, primary hemostasis is sufficient to repair the damage; (3) coagulation phase: The injury of larger vessels induces the exposure of tissue factor, which binds the circulating coagulation factor VII, thus activating the extrinsic pathway of the coagulation cascade and finally leading to thrombin generation [26]. Its primary function is to cleave fibrinogen to fibrin, forming the fibrin clot of the hemostatic plug [28]. This step is named secondary hemostasis [25]; (4) fibrinolytic phase: Once the vascular lesion has been repaired, the clot dissolves through the fibrinolysis process [25]. This lytic activity is mainly performed by plasmin that is generated in situ from its precursor the zymogen plasminogen [29]. This conversion occurs on the surface of the fibrin clot or cell surfaces, through two enzymes, the tissue plasminogen activator or urokinase [29]. 

This process is highly regulated via several mechanisms, including RNA-related pathways. A schematic view of the hemostatic process is shown in Figure 1. 

A perturbation at any step of the hemostatic process may lead to thrombosis and/or bleeding. The modulation of the two faces of hemostasis, such as coagulation cascade and platelet aggregation, is now the cornerstone in the management of thrombotic disorders, such as atrial fibrillation [30], pulmonary embolism [31], acute coronary syndromes [32], peripheral artery diseases [33] and stroke [34]. However, despite the antithrombotic intervention, recurrent thrombotic events still occur, indicating that the mechanisms of the coagulation cascade and platelet activation need to be further investigated. In this regard, in the last 15 years, our understanding of the molecular mechanisms involved in platelet activation has improved, uncovering novel therapeutical strategies to pursue even in antithrombotic interventions. Despite the lack of nuclei, the modulation of platelet genome and proteome has been reported, with important therapeutical implications [17].

## 3. Platelet Genome and Proteome Regulation: Role of miRNA, Splicing mRNA, and Noncoding RNA

### 3.1. miRNome Modulation upon Platelet Activation

The central dogma of molecular biology demonstrates the importance of mRNAs as pivotal mediators between the genetic information at the DNA level and proteomes that regulate the various functional outcomes at the cellular level. For this reason, RNA-Seq and real-time techniques are very used in many different research fields and assume an important role in the study of cardiac development and platelet maturation [35]. It has been largely proved that mRNA levels can be used for research and diagnostical purposes [36], especially if related to cardiovascular diseases [37]. Despite anucleate, platelets possess a repertoire of mRNAs, several of which are translated into protein [38]. These mRNAs may be transferred from platelets to other cells, and these recipients may use the mRNA as a template for translation [38]. Variation in blood mRNA has been reported in ACS patients [35,39] as well as in animal models of acute cerebral ischemia [40,41,42]. In addition, mRNA levels derived from the platelets have been used as indicators of pathological status [35,43,44]. Hence, the characterization of platelet transcriptome may be helpful to obtain further information regarding the function of platelets in health and disease. A summary overview of this process is provided in Figure 2.

A regulation mechanism of gene expression in eukaryotes is controlled through microRNAs (miRNAs), which are small RNAs (21–24 nucleotides) able to regulate the translation of mRNAs by directly binding to them [45]. It has been demonstrated that the activity of the majority of human genes is regulated at the post-transcriptional level through miRNAs [45]. Platelets also possess different miRNAs [46], which are “inherited” from megakaryocytes along with the key components of the “RNA interference” machinery dependent on these small noncoding RNAs [47]. Moreover, during the initial phases of platelet activation, significant modulation occurs in different miRNA levels in the platelets, resulting in significant modifications in the proteome [18,46].

It has been reported that human platelets contain a miRNA repertoire [46,47] and biosynthetic pathway components [48]. The regulatory network existing between platelets’ miRNome, transcriptome, and proteome during activation is a matter of intense investigation. Our group has demonstrated that upon activation, the platelet proteome undergoes significant remodeling [18]. These changes occur in the absence of comparable changes in the steady-state levels of the corresponding mRNAs. Taking into account the known effects of miRNAs on mRNA translation efficiency, we then showed that platelet miRNome underwent major reprogramming under these conditions [18]. This modulation resulted in proteome changes that were independent of the nature of the stimulus [18]. It has also been reported that the extent of the miRNA response is directly correlated with the potency of the activating stimulus. Moreover, it has also been shown that a large number of miRNAs respond to different degrees of platelet activation [46,49,50,51], indicating a strong reorganization of platelet miRNome upon activation [18,52]. This accumulation of miRNAs is the result of an enhanced precursor (pre-miRNA) maturation mediated via the enhanced expression of the key components of miRNA machinery, such as Dicer, GW182, and Ago2 [47,50]. On the other hand, the downmodulation of miRNAs occurs through nucleic acid modifications, such as adenylation and uridylation, which reduce their stability [50], and/or through the selective release of mature miRNAs from activated platelets [49,53,54] via other mechanisms that are still under investigation [21,46]. An inverse correlation between miRNAs and mRNA behavior is also possible, suggesting a putative effect of miRNAs on messenger stability [55]. MiRNAs may also target longer RNAs. This requires their assembly in a RISC complex or microRNP (microRNA–ribonucleic–protein complex (miRNP)), where the miRNA serves as a specificity guide for target RNA recognition [55,56]. The core of every miRNP/RISC includes an Ago protein, which directly binds to a single-stranded miRNA and, upon target RNA recognition, orchestrates mRNA degradation or translational repression [55]. In detail, a perfect complementarity with the target RNA leads to RNA cleavage; an imperfect one, instead, produces translational repression [55]. The final effect is the modulation of protein synthesis, even if the level of the target mRNA remains unchanged [55].

It has been shown that the integrin pathway is one of the most modulated targets by miRNAs in activated platelets [18]. This is not a surprise since upon activation, a substantial change in platelets’ shape occurs [27]. It is well known that integrins play a key role in the adhesion and aggregation of the subendothelial matrix proteins of the vascular wall, thus ensuring hemostasis [57]. Five different integrins, belonging to the β1 and β3 families, (α2β1, α5β1, α6β1, αvβ3, and αIIbβ3, whose main ligands are collagen, fibronectin, laminins, vitronectin, and fibrinogen, respectively) are expressed at platelet surface [57]. The most abundant and best-characterized integrin is αIIbβ3 [58]. Recently, the importance of α5β1 in hemostasis under normal and inflammatory conditions has also been better defined [59]. Various agonists may modulate the affinity of integrins for their ligands, thereby reinforcing platelet activation [57]. Integrins’ expression, as well as the intracellular integrin-related pathway, are highly regulated [60]. Specifically, the upregulation of miR-92b-3p, miR-486–3p, and let7-e-5p has been reported [18,46,49,50]. 

Additionally, miR-107 and miR-15b-5p are reported to be downregulated in platelets following activation [18,50]. More interestingly, upon activation, several platelet miRNAs targeting the immunoinflammatory response pathway are modulated [18]. Several other miRNAs have been linked to platelet activity and aggregation [46]. We will discuss the most significant ones.

One of the most abundant miRNAs present in platelets is miR-223 [61]. It is involved in the regulation of the mRNA of the P2Y12 receptor, one of the main receptors involved in aggregation [16] and pharmacologically modulated in most antithrombotic strategies [62]. Thus, a putative role as a biomarker of platelet reactivity has been proposed and investigated for miR-223 [61]. Moreover, it has been reported that thrombin-activated platelets may release the complex Ago2/miR-223 [63], and this complex may be internalized via endothelial cells, thus modifying gene and protein expression [63]. Different reports support the prognostic role of plasma circulating miR-223 in the incidence of myocardial infarction with 10 years of follow-up [64], as well as the diagnostic value in healthy subjects treated with antiplatelet agents [65] and ACS patients [66]. Another highly expressed platelet miRNA involved in the modulation of P2Y12 receptor expression is miR-126 [67,68]. It also affects the expression of proteins involved in platelet adhesion [69]. Moreover, its role in platelet reactivity has also been reported [70]. Finally, miR-126 modulates the proteins encoded by the PLXNB2 gene, which belongs to the receptors for semaphorins involved in platelets’ actin dynamics and thrombus formation [70].

The in vitro activation of platelets by arachidonic acid induces the release of miR-126 [71,72], which is also inhibited by acetylsalicylic acid (ASA) [73]. In diabetic patients, a correlation between circulating miR-126 levels and platelet reactivity measured with P-selectin was found [74]. The administration of ASA in this group was associated with a decrease in circulating miR-126 [74]. Based on the available data, the final effect of miR-126 on platelet aggregation results from the modulation of P2Y12 receptor activation plus platelet adhesion to collagen [75]; thus, the miR-126 downregulation associated with ASA administration impaired platelet activity [67].

In the last few years, special attention has also been paid to the miR-19b-1-5p that is associated with thromboxane-mediated platelet aggregation [46]. This miRNA belongs to the miR-19b cluster that has been reported to have antithrombotic properties [76]. A previous in vitro study indicates that miR-19b expression inhibits the expression of the endothelial tissue factor and its procoagulant activity [77]. Taking into account these properties, the downregulation of miR-19b-1-5p may potentially be associated with increased platelet reactivity [46]. Two reports have shown that decreased levels of miR-19b-1-5p in isolated platelet samples after ASA administration are associated with sustained platelet aggregation in healthy subjects [78] and higher risk of major cardiovascular events in ACS patients [79], supporting its putative role as a biomarker to monitor antiplatelet therapy.

Another miRNA that has been linked to increased platelet reactivity is miR-204-5p [80]. Its regulation seems to occur through CDC42 downregulation and the modulation of fibrinogen receptor expression (αIIbβ3) [81]. CDC42 is actively involved in cytoskeleton dynamics, and it is a direct target of miR-204-5p [82]. A significant upregulation of miR-204-5p has been reported in ACS patients on dual antiplatelet therapy (ASA plus clopidogrel), showing high platelet reactivity [83]. 

A pool of other miRNAs, specifically miR-15a, miR-339-3p, miR-365, miR-495, miR-98, and miR-361-3p, are linked to the mTOR signaling, which is known to correlate with glycoprotein-VI-mediated platelet aggregation [84].

### 3.2. Role of the mRNA Splicing

Immature RNAs represent a significant fraction of platelet transcriptome [23]. These RNAs contain one or more introns because of the ‘alternative splicing’ in megakaryocytes and are therefore incompetent for protein synthesis [85,86]. During platelet activation, a maturation process of these mRNAs is triggered, which leads to an increase in the synthesis of different regulatory proteins of key processes in the platelet, as demonstrated by in-depth proteogenomic analyses [18,52,87,88]. Understanding proteome changes following platelet activation is a new way to diagnose, monitor, and treat diseases caused by platelet dysfunction [52,88]. It has been shown that platelet proteome can be reorganized through post-transcriptional/translational processes [87]. In this regard, cytoplasmic splicing may be a possible regulatory mechanism active during platelet activation [89,90]. The role of intron retention (IR)-induced mRNA modulation is a relevant process, which is active during hemopoietic lineage maturation [91] and platelet activation [87]. It is known that IR is usually associated with mRNA degradation [91]. However, some reports indicated that it could also influence protein production. It has been demonstrated that transcripts carrying retained introns are present in megakaryocytes and anucleate platelets [91]. These introns may be spliced out upon platelet activation [87]. Studies combining proteome and transcriptome profiling data from resting and activated platelets have definitely shed more light on the relationship between mRNAs and proteins in human platelets. Activation may lead to the modulation of thousands of proteins from ~8000 protein-coding RNAs detected [18,87]. On the other hand, IR analysis revealed a very significant number of immature RNAs, many of them being matured during the activation process [87,92]. The induction of resident pre-mRNA maturation during platelet activation promotes selective changes in the platelet proteome through the neo-synthesis of proteins involved in platelet shape changes and possibly other key processes during thrombosis [87].

In conclusion, together with other several processes, such as the modulation of miRNA expression [18,21,46,49,54], the extensive maturation of resident pre-mRNAs occurs in platelets in response to activating stimuli [87], representing a mechanism for the post-transcriptional control of proteome composition in these anucleated cell fragments. 

Moreover, some of the pathways involved in platelet response to activating stimuli [93] overlap with the mechanisms involved in splicing control, thus being amenable to pharmacological modulation [94].

### 3.3. Involvement of lncRNA in Platelet Function

Long noncoding RNAs (lncRNAs) are long RNA transcripts (usually more than 200 nucleotides) not translated into proteins [95]. Their main role is the modulation of the crucial functions of other noncoding RNAs such as miRNAs, small nucleolar RNAs (snoRNAs), etc. [96,97]. It is known that lncRNAs originate from their own promoters, the promoters of coding or noncoding genes, or enhancer sequences [96,98]. lncRNAs are present in many organisms with a higher degree of tissue specificity [99] and distinctive evolutionarily conserved patterns [99,100,101]. It has been reported that lncRNAs affect a wide range of cellular activities and functions in health and diseases such as cancer [102], neurological [103] and cardiovascular [23] conditions, and immunological and metabolic disorders [104,105]. Notably, lncRNAs may (a) activate or repress gene expression through the relocalization of regulatory factors; (b) aid in the formation of ribonucleoprotein (RNP) complexes; (c) remove the regulatory factor bound to the genome, thereby terminating its regulation; (d) inhibit the miRNA-mediated gene repression; (e) function as primary miRNA precursors that are processed into mature miRNAs; and (f) initiate long-range gene regulation [95,96,97,105].

Long noncoding RNAs seem to be highly involved in megakaryocyte development and platelet production [106]. It has also been shown that a large number of lncRNAs are present in platelets despite the lack of nuclei, suggesting that epigenetic regulation may be an important regulatory mechanism to modulate the platelet proteome and adapt to environmental situations over their lifespan [23,96]. In the absence of nuclei, post-transcriptional mechanisms are the key systems for gene expression regulation mainly via noncoding RNAs [96,98]. Accumulated evidence indicates that lncRNAs play an important role in platelet reactivity [107]. In this regard, it has been reported that lncRNA ENST00000433442 is significantly correlated with high platelet reactivity [108], while the knockdown of lncRNA metallothionein 1 pseudogene 3 (MT1P3) may inhibit platelet activation [68], indicating a potential link between platelet lncRNAs and platelet functions. Specifically, lncRNAs exert a different expression profile between hyperreactive and hyporeactive platelets [107]. The multiple genes/signaling pathways associated with platelet functions are influenced by differentially expressed platelet lncRNAs [23,107,109]. Additionally, other genes/pathways linked to platelet-mediated effects on other cells/tissues are also influenced by these differentially expressed platelet lncRNAs [96], presenting a further scenario to be explored. Thus, lncRNAs in circulating platelets might represent a novel potential biomarker or a possible therapeutic target in many diseases. To date, four lncRNAs (LNCAROD, SNHG20, LINC00534, and TSPOAP-AS1) have been reported to be upregulated in platelets of colorectal cancer (CRC) patients and are potential biomarkers for CRC diagnostics [110]. Further evidence links lncRNAs with cardiovascular diseases [111]. Specifically, the lncRNA ENSG00000258689 is downregulated in hyper-reactive platelets, in patients with AMI [107]. Notably, the aberrant expression of ENSG00000258689 in AMI patients could be partially reversed with the use of aspirin [107]. Moreover, it has also been shown that the lncRNA ENST00000433442 is an independent risk factor for high residual platelet reactivity in patients affected by ischemic heart diseases already under dual antiplatelet therapy [108], further supporting the role of lncRNA involvement in platelet reactivity.

### 3.4. Other Noncoding RNAs (Small Nucleolar RNAs, Y-RNAs, Circular RNAs, and piRNA): Is There a Role in Platelet Activity?

In the last few years, other noncoding RNAs have been identified, such as small nucleolar RNAs, Y-RNAs, and circular RNAs, opening a new field of investigation concerning their role in health and diseases [112]. Small nucleolar RNAs (snoRNAs) are small noncoding RNAs (between 60 and 200 nt) found in the nucleolus, mainly encoded by intronic regions of both protein coding and non-protein coding genes [113]. Their primary function is the 2′-O-methylation and pseudouridylation of rRNAs (ribosomal RNAs) [113]. Is also postulated that snoRNAs might regulate cell physiology by guiding N4-acetylcytidine (ac4C) modifications, modulating alternative splicing, and performing miRNA-like functions, thus enabling protein synthesis [113]. It has been reported that the plasma levels of snoRNAs (mainly SNORD113.2 and SNORD114.1) correlate with platelet activation [114], supporting their putative role in platelet function.

Y-RNAs are small noncoding RNAs of approximately 100 ± 20 nucleotides in size, involved in several cellular processes, including DNA replication, RNA stability, and cellular stress responses [115]. They fold into characteristic stem–loop secondary structures that include a loop domain, an upper and lower stem domain, and a polyuridine domain [115]. The lower stem domain as well as the polyuridine tail are highly conserved binding sites for Ro60 and La proteins, respectively, and are essential for Y-RNAs to associate with these proteins to form RoRNPs [115]. Ro60 ribonucleoprotein particles are necessary for DNA replication through interactions with chromatin and initiation proteins [116]. Although Y-RNAs are similar in size to miRNAs, it has been shown that these RNA fragments are not involved in the microRNA pathway [117]. However, the levels of both platelet-derived plasma miRNAs and Y-RNAs have been linked to platelet function [67]. In this regard, a previous study reported a correlation between miRNAs and Y-RNA fragments with platelet activation markers in the general population from the Bruneck Study. Furthermore, plasma miRNA and Y-RNA levels were associated with residual platelet activity in ACS patients on dual antiplatelet therapy [67]. Furthermore, a recent review underlined a possible role of Y-RNAs in atherosclerosis through the release in the bloodstream of extracellular vesicle (EV)-enclosed Y-RNAs [118].

piRNAs were first discovered in the testes of Drosophila melanogaster [119], and they were later associated with transposable elements [120]. Today, it is known that piRNAs are small noncoding RNAs of 26–32 nucleotides and are quite conserved among species [121]. They interact with the PIWI proteins of the AGO family (from this derives the name Piwi-interacting RNAs (piRNAs)), forming a silencing complex able to suppress transposable elements and regulate gene expression at both epigenetic and post-transcriptional levels [121,122]. It has been demonstrated that this class of molecules can have an important role in cardiovascular diseases. Indeed, they seem to be involved in heart failure, myocardial infarction, angiogenesis, and ischemic damage [123]. Furthermore, many studies have shown that piRNAs are involved in regulating many factors such as cell proliferation, apoptosis, cell cycle, cell migration, oxidative stress, and DNA damage [123]. Recently, it has been hypothesized that piRNAs are not easily degraded and can pass through the cell membrane [124]. Indeed, they were also proposed as a tool to cloak the platelet membrane with nanoparticles and thus increase the permeability of the blood–brain barrier to treat neurological cancers [125]. To date, the role of piRNAs in platelet activity is not completely understood, but it is known that they may synergize with miRNAs to promote megakaryocyte differentiation [126].

Circular RNAs (or circRNAs) are single-stranded RNAs that form a covalently closed continuous loop [112,127], with the 3’ and 5’ ends joined together [128]. This feature confers numerous properties to circRNAs, many of which have only recently been identified. The biological function of most circRNAs remains unclear; however, because they are resistant to degradation, a putative role in mRNA stability in the absence of transcription is postulated [23,129]. The identification of circRNAs’ mechanism of action is very challenging and includes the characterization of alterations in the host gene transcript (particularly for nuclear circRNAs), the identification of the interactions with RNAs, and the assessment of the circRNA–protein relationship [112]. It has been reported that some circRNAs may code for proteins [130], while others have shown potential as gene regulators [127,128]. CircRNAs are highly abundant in human platelets [129]. To date, no evidence is available on the role of circRNAs in platelet function. A summary of noncoding RNAs and their involvement in platelet function are provided in Table 1.

An overview of noncoding RNAs in platelet is provided in Figure 3.

## 4. Potential Clinical Implications and Future Perspectives: Are We Ready for Daily Use of Noncoding RNAs in Clinical Practice?

In the last few years, technological innovation has led to a great improvement in analyzing cellular functions. 

The classic view of genome translation from DNA to proteins has deeply changed even in cells that lack nuclei such as platelets. It is now well known that these cell fragments are not only the final effectors of the coagulation cascade but also can participate in several other biological functions beyond thrombosis [17,19]. Upon activation, platelets are able to modulate their own megakaryocyte-derived transcripts that finally lead to proteome change [17,85,86]. This modulation is the final result of complex pathways that are still under intense investigation. New discoveries are focused on the role of miRNAs, mRNA alternative splicing, and noncoding RNAs [24]. 

The possibility to monitor the changes in mRNA and miRNA profiles, detect the alternative mRNA transcripts, and evaluate the different noncoding RNAs (lncRNAs, Y-RNAs, and circRNAs) could unveil new interesting perspectives in clinical practice. It is known that platelets may secrete over 500 different molecules into the plasma, including a large number of noncoding RNAs and miRNAs [24].

MicroRNAs are abundant in platelets [18,46]. They may regulate platelet function by targeting specific genes, thus modifying protein expression [18]. Selected platelet-derived miRNAs have been linked to platelet reactivity. Hence, they might be useful diagnostic and prognostic biomarkers of high on-treatment platelet reactivity. To date, many authors have sought to study transcriptomic data derived from human platelets [35] and their relationship with microRNA data [131]. These data, collected by many authors over the years, underline the remarkable features of the transcript–protein network in platelets. However, these features are still not completely known. In particular, not only the data collected on healthy donors [132] but also the data collected during acute myocardial infarction [35], cancerous events [133], and other pathological events [86,134] are extremely useful to understand platelet behavior (activation, aggregation, and proteomic turnover on the surface). Furthermore, over the years, many methods have been exploited to analyze platelet transcriptomes: bulk RNA, single-cell RNA, and gene expression arrays [85]. However, to date, single-cell data are still limited due to the very small number of RNAs contained within the platelets [85]. Thus, further investigation is needed to better identify their clinical application. 

The splicing variants of mRNAs are also interesting and might be of clinical relevance [90,94]. Alternative splicing occurs in platelets, especially in young platelets that are enriched with prothrombotic signaling. This prothrombotic potential is abundant in patients with diabetes, acute or chronic coronary syndrome [135], and smokers; thus, it might be evaluated for therapeutical purposes. 

Finally, alterations in the repertoire and/or the number of platelet-secreted noncoding RNAs have been associated with CVD as well as other diseases [23].

The miRNA-based assessment of platelet reactivity, as well as the evaluation of mRNA alternative transcripts or noncoding RNA profiles, may improve the prediction of antiplatelet treatment efficacy, giving the opportunity of an individual antiplatelet treatment tailored to highly specific patient needs. 

The analysis of these data has allowed researchers to focus their attention on a few pathways or genes dysregulated during pathological events (such as acute myocardial infarction [35]); this could be very useful to directly test the expression of the selected genes using easier and more cost-effective approaches such as real-time methods (now available in many hospitals). However, many limitations remain, due to the biological conformation of platelets (platelets’ size and RNA contents), the biological variance between individuals [131], and technical difficulties (e.g., accidental activation during sample collection).

## 5. Conclusions

The role of noncoding RNAs is essential for the modulation of platelet function. Undoubtedly, RNA-based diagnostics are promising tools to monitor platelet activity, CVD, and other diseases [136]. The advantages of RNAs as biomarkers include the possibility of detection in human fluids, a cell-type specific profile, and fluctuations in response to stimuli. However, some technical challenges still exist: (1) there is a need for a higher amount of blood (at least 15/20 mL), and (2) their time-consuming analysis requires several hours to obtain the results, thus limiting their application in acute settings. Focusing on a few dysregulated genes/noncoding RNAs might be a fair compromise to pursue.

RNAs may be also administered for therapeutical purposes. The recent approval of RNA-based drugs to treat some CVD such as hypercholesterolemia (i.e., inclisiran) [137] as well as other diseases [138] indicates that a new therapeutical future is on the horizon. However, there is still a need for further technological improvements for the best use of this approach by overcoming the current limitations. First, despite the recent advances, the issue of the immunogenicity of RNA therapeutics still needs to be further explored. Second, the specificity, with unexpected off-target effects and undesired on-target effects, as well as the appropriate relationship between dosing and specificity still needs to be better defined in the clinical setting. Finally, “safe” delivery to the target organ with more selective and specific approaches should be achieved.

## Figures and Tables

**Figure 1 ijms-24-07650-f001:**
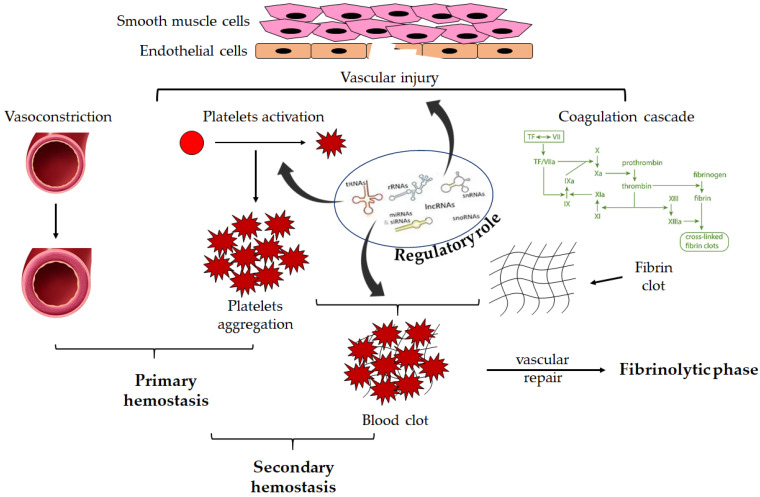
Overview of hemostasis. Endothelial damage induces activation of the primary hemostasis. Subendothelial thrombogenic material is exposed to the flowing blood. Vasoconstriction and coagulation cascade activation occur. Moreover, the subendothelial matrix proteins bind to receptors on the platelet surface finally resulting in platelet activation and aggregation, leading to platelet plug formation. Secondary hemostasis leads to the formation of fibrin through coagulation proteins and the formation of a blood clot including activated platelets. Once the vessel wall is repaired, the clot is dissolved by fibrinolysis. These processes are regulated via different RNA-related mechanisms.

**Figure 2 ijms-24-07650-f002:**
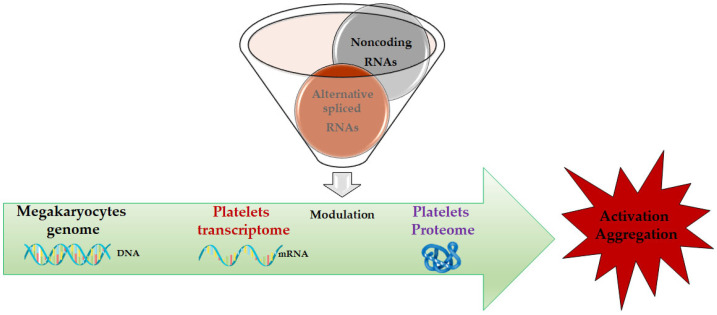
The central dogma of biology in platelets: from megakaryocyte genome to platelet proteome via platelet transcriptome modulation. The focus is on noncoding RNAs and alternatively spliced mRNAs.

**Figure 3 ijms-24-07650-f003:**
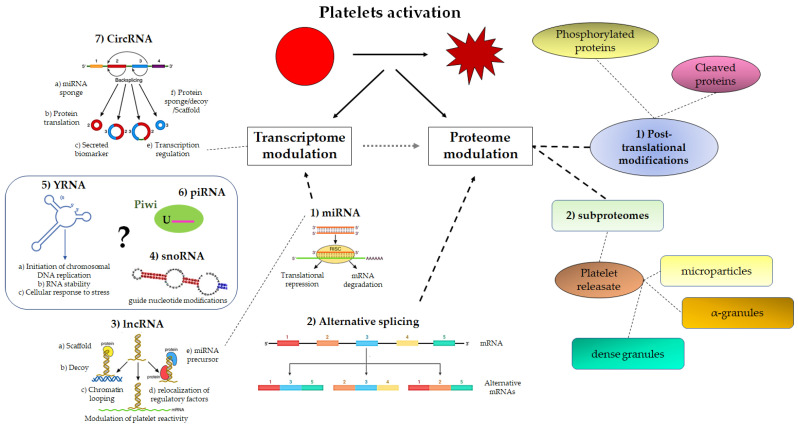
Schematic view of platelet transcriptome/proteome modulation upon activation. This diagram illustrates the complex interplay between platelets’ transcriptome and proteome via miRNAs and mRNA alternative splicing. It is also reported that noncoding RNAs might affect the transcriptome (see text for details). Finally, post-translational modifications may occur once platelet proteins are synthesized. MiRNA: microRNA; lncRNA: long-noncoding RNA; snoRNA: small-nucleolar RNAs; circRNA: circular RNA; piRNA: piwi RNA.

**Table 1 ijms-24-07650-t001:** RNA-related mechanisms involved in platelet function.

RNA-Related Activity	Nucleotides	Function	Relevance
miRNA	~22	Downregulation of protein synthesis	Most studied platelet ncRNAs
Alternatively spliced mRNA		Different exons combinations from the same gene, leading to different, but related, mRNA transcripts	An important mechanism in anucleate cells with prespecified transcriptome to modulate protein synthesis
lncRNA	>200	Role in platelet reactivity	Highly involved in megakaryocytes development and platelet production
snoRNA	60–200	Possible role in platelet activation	The primary function is to guide nucleotide modifications in rRNA
Y-RNA	100 ± 20	Possible role in platelet function	Involvement DNA replication, RNA stability, and cellular stress responses
piRNA	26–32	Possible synergism with miRNAs to promote megakaryocyte differentiation	A silencing complex able to suppress transposable elements
circRNA	single-stranded RNA that forms a covalently closed continuous loop	Highly abundant in human platelets with a not-well-defined function	Putative role in mRNA stability

## Data Availability

Not applicable.

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
