# Peer review of "The Novel Role of Noncoding RNAs in Modulating Platelet Function: Implications in Activation and Aggregation"

_ijms, 2023, doi:10.3390/ijms24087650_

Round 1
Reviewer 1 Report
Cimmino G. et al provide an overview of molecular mechanisms involved in the modulation of platelet functioning, with a main focus on the complex interplay between the platelet transcriptome and proteome via different RNAs. The aim of this review is to offer a better understanding of the role of these RNAs in platelet activation and potential roles in diagnostics and therapy. The authors selected indeed an interesting and relevant topic to review, which fits into the scope of this journal, and gave an extensive overview on of the involvement of the different RNAs (miRNA, mRNA, lnrRNA and other noncoding RNAs) in platelet functioning. Hereby, they highlight their potential as biomarker (eg. in diabetes and CVD), and as therapeutical tool (RNA-based drugs to treat CVD). To make this overview complete, the authors also explain the technical issues concerning RNA-based diagnostics. Overall, the paper is comprehensive. However, there are many typo’s and in some parts of the paper the English is poor and needs corrections.
I support the publication of this paper, after minor corrections.
Minor comments
- The authors cite a lot of reviews while it would be better to cite the original work.
- The title ‘Novel insights in molecular mechanisms’ is too broad, it would be good to specify the title a bit.
- Page 4, line 164-166:Please specify the involvement of integrin signalling in platelet shape change.
- Page 5, line 172-175: Please rephrase this sentence.
Do you mean P2Y12 is an mRNA receptor? Not clear.
When mentioning P2Y12, explain that P2Y12 inhibitors are used in the clinic. This is explained on line 208, but should come earlier.
- Page 5, line 196-198: You mention ‘studies’, but only one reference is given. Are there more studies?
- Page 5, line 206: With fibrinogen receptor expression, do you mean integrin αIIbβ3? Also GPVI has been shown to be a receptor for fibrinogen so be more specific.
- Page 7, line 307-309: Please specify: how is this different in ACS patients compared to (healthy?) general population? Why are you making this subdivision? What does this imply?
Improve English:
- Page 1, line 16: Platelets are a major player à Platelets are major players
- Page 1, line 20: our acknowledgement had improved à you mean ‘our knowledge’?
- Page 1, line 27: will open new scenario à will open new perspectives
- Page 1, line 30: Implication à implications
- Page 2, line 60: Associated to à associated with
- Page 2, line 89-90: platelets released different factors their own granules à platelets released different factors from their own granules
- Page 2, line 91: Occurà occurs
- Page 3, line 112: Ledà lead
- Page 3, line 117: Eventà events
- Page 3, line 119: Plateletsà platelet
- Page 4, line 130: Possess à contain
- Page 4, line 135: Platelets contains miRNA repertoire à Platelets contain a miRNA repertoire
- Page 4, line 136: Between platelet miRNome à ‘the platelet’
- Page 4, line 138: Platelet à the platelet
- Page 4, line 143: A proteome changes à proteome changes
- Page 4, line 164: Target à targets
- Page 4, line 170: Hasà have
- Page 5, line 184: Implicating in à involved in?
- Page 5, line 191: P2Y12 receptor à P2Y12 receptor ‘stimulation/activation’? Sentence is not complete.
- Page 5, line 212: Glycoprotein mediated à Glycoprotein VI?
- Page 6, line 232: Ledà lead
- Page 6, line 242: Sentence is not very clear. Perhaps an example of a pathway involved in both processes, and pharmacological modulation?
- Page 6, line 266:isàare and Systemà systems
- Page 7, line 323: Hypothesize à hypothesized
- Page 8, line 326: Bad construction
- Page 8, line 339: Are à is?
- Page 9, line 360: Scenario à perspectives?
Author Response
We thank the reviewer for the time he/she spent to review our article and for the criticisms they raised. We felt they were appropriate and the revised version of our manuscript is improved because of that. To facilitate the readers, all changes are in red.
Minor comments
- The authors cite a lot of reviews while it would be better to cite the original work.
ANSWER. We thank the reviewer for his/her suggestion. We have added original work to existing references
- The title ‘Novel insights in molecular mechanisms’ is too broad, it would be good to specify the title a bit.
ANSWER. We thank the reviewer for his/her suggestion. Title has been rephrased focusing on the main topic of the review
- Page 4, line 164-166: Please specify the involvement of integrin signalling in platelet shape change.
ANSWER. We thank the reviewer for his/her suggestion. Involvement of integrin signalling in platelet has added (page 4, line 166-174)
- Page 5, line 172-175: Please rephrase this sentence.
Do you mean P2Y12 is an mRNA receptor? Not clear.
When mentioning P2Y12, explain that P2Y12 inhibitors are used in the clinic. This is explained on line 208, but should come earlier.
ANSWER. We thank the reviewer for his/her suggestion. The sentences have been rephrased as indicated
- Page 5, line 196-198: You mention ‘studies’, but only one reference is given. Are there more studies?
ANSWER. We apologize for the wrong information. One study is available on this topic
- Page 5, line 206: With fibrinogen receptor expression, do you mean integrin αIIbβ3? Also GPVI has been shown to be a receptor for fibrinogen so be more specific.
ANSWER. We apologize for the misunderstand. We have specified αIIbβ3
- Page 7, line 307-309: Please specify: how is this different in ACS patients compared to (healthy?) general population? Why are you making this subdivision? What does this imply?
ANSWER. We apologize for the confusing information. We have reported the study from Kaudewitz D. et al (Circulation Research. 2016;118:420–432). In this study, miRNA measurements were performed in 669 subjects of a population-based study (Bruneck Study), as well as in 125 patients with ACS. The study correlates miRNAs with platelet activation markers in the general population and with the residual platelet activity in patients with ACS on antiplatelet therapy. Most but not all abundant platelet miRNAs were positively correlated with the VerifyNow P2Y12 and VASP assays, which are standardized assays for assessing the effects of P2Y12 inhibitors. In addition it has been reported an association of plasma YRNAs with platelets and that inhibition of miR-126 attenuates platelet aggregation in response to low but not high agonist concentrations. Thus, we have better specified this information in the main manuscript to indicate that YRNAs might be a possible marker in cardiovascular diseases.
Improve English:
- Page 1, line 16: Platelets are a major player à Platelets are major players
- Page 1, line 20: our acknowledgement had improved à you mean ‘our knowledge’?
- Page 1, line 27: will open new scenario à will open new perspectives
- Page 1, line 30: Implication à implications
- Page 2, line 60: Associated to à associated with
- Page 2, line 89-90: platelets released different factors their own granules à platelets released different factors from their own granules
- Page 2, line 91: Occurà occurs
- Page 3, line 112: Ledà lead
- Page 3, line 117: Eventà events
- Page 3, line 119: Plateletsà platelet
- Page 4, line 130: Possess à contain
- Page 4, line 135: Platelets contains miRNA repertoire à Platelets contain a miRNA repertoire
- Page 4, line 136: Between platelet miRNome à ‘the platelet’
- Page 4, line 138: Platelet à the platelet
- Page 4, line 143: A proteome changes à proteome changes
- Page 4, line 164: Target à targets
- Page 4, line 170: Hasà have
- Page 5, line 184: Implicating in à involved in?
- Page 5, line 191: P2Y12 receptor à P2Y12 receptor ‘stimulation/activation’? Sentence is not complete.
- Page 5, line 212: Glycoprotein mediated à Glycoprotein VI?
- Page 6, line 232: Ledà lead
- Page 6, line 242: Sentence is not very clear. Perhaps an example of a pathway involved in both processes, and pharmacological modulation?
- Page 6, line 266:isàare and Systemà systems
- Page 7, line 323: Hypothesize à hypothesized
- Page 8, line 326: Bad construction
- Page 8, line 339: Are à is?
- Page 9, line 360: Scenario à perspectives?
ANSWER. We apologize for the typos. We have extensively rechecked the manuscript and modified accordingly
Reviewer 2 Report
Authors reviewed the novel insight in molecular mechanisms modulating platelets function: implications in activation and aggregation and the emerging regulatory pathways involved in platelet activation and aggregation was discussed.
Although this manuscript is potentially interesting, several issues arise.
Figure 1 is not attractive and has no new finding.
Figure 2 many abbreviations should be explained.
Additional figures and tables may be helpful for the readers to understand your review.
There were no data for platelet activation or aggregation.
The relationship between mRNA and platelet activation or aggregation may be required.
Clinical application of your work may be helpful.
How do we measure or evaluate mRNA in acute coronary diseases or acute cerebral infarction?
The behavior of mRNA in circulation or platelet may be helpful in acute coronary diseases or acute cerebral infarction.
Conclusion needs take home message.
Author Response
We thank the reviewer for the time he/she spent to review our article and for the criticisms they raised. We felt they were appropriate and the revised version of our manuscript is improved because of that. To facilitate the readers, all changes are in red.
Although this manuscript is potentially interesting, several issues arise.
Figure 1 is not attractive and has no new finding.
ANSWER. We thank the reviewer for his/her criticism. We have improved the figure 1 with further novel details to be more attractive
Figure 2 many abbreviations should be explained.
ANSWER. We thank the reviewer for his/her suggestion. We have improved Figure 2 and specified the abbreviation in the figure legend
Additional figures and tables may be helpful for the readers to understand your review.
ANSWER. We thank the reviewer for his/her suggestion. We have improved the two figures and added a table explaining the role of noncoding RNA in platelets
There were no data for platelet activation or aggregation.
ANSWER. With the respect for the reviewer, we would like to point out that in each paragraph there is a mention on the relationship with platelet function (activation/reactivity) .
The relationship between mRNA and platelet activation or aggregation may be required.
ANSWER. We thank the reviewer to point this out. Characterization of platelet mRNAs has provided a wealth of information regarding the function of platelets in health and disease. We have added this information in page 4, lines 127-130
Clinical application of your work may be helpful.
ANSWER. This is a critical issue of the present manuscript that is provided in section 4
How do we measure or evaluate mRNA in acute coronary diseases or acute cerebral infarction?
ANSWER. We thank the reviewer for his/her suggestion. It has been reported that evaluation of mRNA during cardiovascular event can be performed using Next Generation Sequencing. We have added this information to the revised version of the manuscript (page 4, lines 131-132)
The behavior of mRNA in circulation or platelet may be helpful in acute coronary diseases or acute cerebral infarction.
ANSWER. We thank the reviewer for his/her suggestion. As for the previous criticism, we have added this information to the revised version of the manuscript (page 4, lines 131-132)
Conclusion needs take home message.
ANSWER. We thank the reviewer for his/her suggestion. A final message has been added
Round 2
Reviewer 2 Report
Thank you for your responses.
I am not still cleat the relationship between pathological state and mRNA.
Some table may be helpful.
Abstract and conclusion are required to be clear.
Author Response
Dear reviewer,
thank you for your comments
We have expanded the role of mRNA in our manuscript (page 4, lines 127-133) and added a new schematic figure to help the readers in understandig the process from megakariocytes genome to platelets proteome. However, we would like to point out that mRNA is not in the main scope of the manuscript since we are exploring the novel role of noncoding RNAs.
Abstract and conclusions are now claerly stating the scope and the main message of the manuscript
Thank you for your consideration